

# KL-MOB: automated COVID-19 recognition using a novel approach based on image enhancement and a modified MobileNet CNN

Mundher Mohammed Taresh[1], Ningbo Zhu[1], Talal Ahmed Ali Ali[1], Mohammed Alghaili[1], Asaad Shakir Hameed[2] and Modhi Lafta Mutar[2]

[1] College of Information Science and Engineering, Hunan University, Changsha, Hunan, China
[2] Department of Mathematics, General Directorate of Thi-Qar Education, Ministry of Education, Thi-Qar, Iraq

Corresponding author
Ningbo Zhu, quietwave@hnu.edu.cn

## ABSTRACT

The emergence of the novel coronavirus pneumonia (COVID-19) pandemic at the end of 2019 led to worldwide chaos. However, the world breathed a sigh of relief when a few countries announced the development of a vaccine and gradually began to distribute it. Nevertheless, the emergence of another wave of this pandemic returned us to the starting point. At present, early detection of infected people is the paramount concern of both specialists and health researchers. This paper proposes a method to detect infected patients through chest x-ray images by using the large dataset available online for COVID-19 (COVIDx), which consists of 2128 X-ray images of COVID-19 cases, 8,066 normal cases, and 5,575 cases of pneumonia. A hybrid algorithm is applied to improve image quality before undertaking neural network training. This algorithm combines two different noise-reduction filters in the image, followed by a contrast enhancement algorithm. To detect COVID-19, we propose a novel convolution neural network (CNN) architecture called KL-MOB (COVID-19 detection network based on the MobileNet structure). The performance of KL-MOB is boosted by adding the Kullback–Leibler (KL) divergence loss function when trained from scratch. The KL divergence loss function is adopted for content-based image retrieval and fine-grained classification to improve the quality of image representation. The results are impressive: the overall benchmark accuracy, sensitivity, specificity, and precision are 98.7%, 98.32%, 98.82% and 98.37%, respectively. These promising results should help other researchers develop innovative methods to aid specialists. The tremendous potential of the method proposed herein can also be used to detect COVID-19 quickly and safely in patients throughout the world.

## INTRODUCTION

The novel coronavirus 2019 (COVID-19) is a recently recognized disease caused by the severe acute respiratory syndrome coronavirus 2 (SARS-CoV-2). Being highly

transmissible and life-threatening, it has rapidly turned into a global pandemic, affecting worldwide health and well-being. Tragically, no effective treatment has yet been approved for patients with COVID-19. However, patients can have a good chance of survival if they are diagnosed sufficiently early, where they would undergo the plan of remedial measures correctly.

As a widely available, time- and cost-effective diagnostic tool, chest x-rays (CXRs) can potentially be used for early recognition of COVID-19. Nevertheless, COVID-19 can share similar radiographic features with other types of pneumonia, making it difficult for radiologists to manually distinguish between the two. As a result, manual detection of COVID-19 is time-consuming and mistake-prone because it is left to the subjective judgment of the radiologist. It is thus highly desirable to develop automated detection techniques.

With the rapid global spread of COVID-19, researchers have begun using state-of-the-art deep-learning techniques to automate the recognition of COVID-19. The initial lack of COVID-19 data compelled earlier researchers to use pretrained networks to build their own models (*Narin, Kaya & Pamuk, 2020*; *Ozturk et al., 2020*; *Apostolopoulos & Mpesiana, 2020*; *Civit-Masot et al., 2020*; *Albahli, 2020*; *Sethy & Behera, 2020*; *Apostolopoulos, Aznaouridis & Tzani, 2020*; *Chowdhury et al., 2020*; *Farooq & Hafeez, 2020*; *Maghdid et al., 2020*; *Hemdan, Shouman & Karar, 2020*; *Taresh et al., 2021*; *Punn & Agarwal, 2021*). Given that COVID-19 infected millions of people worldwide within a few months of its detection, a mid-range dataset of positive cases was made available for public use (*Wang, Lin & Wong, 2020*). This dataset can be uploaded from https://github.com/lindawangg/COVID-Net/blob/master/docs/COVIDx.md. This, in turn, has enabled further progress in developing new, accurate, in-depth models for COVID-19 recognition (*Ahmed et al., 2020*; *Afshar et al., 2020*; *Ucar & Korkmaz, 2020*; *Luz et al., 2020*; *Hirano, Koga & Takemoto, 2020*; *Rezaul Karim et al., 2020*). However, some medical imaging issues usually pose difficulties in the recognition task, reducing the performance of these models. These issues include, but are not limited to, insufficient training data, inter-class ambiguity, intra-class variation, and visible noise. These problems oblige us to significantly enhance the discrimination capability of the associated model. Specifically, regarding the x-ray image, the common characteristics are grayscale color space, high noise, low intensity, poor contrast, and weak boundary representation, which will normally affect the information of the image (*Ikhsan et al., 2014*).

One way around these issues is to use proper image preprocessing techniques for noise reduction and contrast enhancement. A closer look at the available images reveals the presence of various types of noise, such as impulsive, Poisson, speckle, and Gaussian noise (see Fig. 1 for the most common types of noise in x-ray images (*Paul, Perumal & Rajasekaran, 2018*)). However, the most prevalent studies have focused only on some of these types of noise (*e.g.*, Gaussian and Poisson). In particular, among many other techniques, histogram equalization (HE) (*Civit-Masot et al., 2020*; *Tartaglione et al., 2020*, *Rezaul Karim et al., 2020*), contrast limited adaptive histogram equalization (CLAHE) (*El-bana, Al-Kabbany & Sharkas, 2020*; *Saiz & Barandiaran, 2020*; *Maguolo & Nanni, 2021*; *Ramadhan et al., 2020*), adaptive total variation method (ATV) (*Punn & Agarwal,*

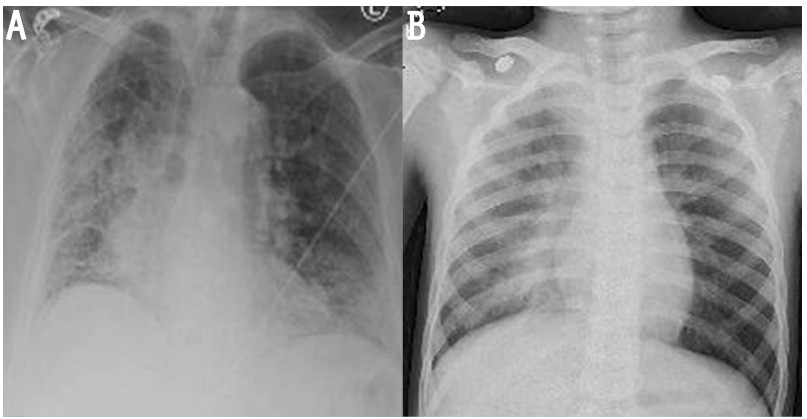

**Figure 1 Noisy images: (A) image with impulsive noise and (B) image with Gaussian noise.**

*2021*), white balance followed by CLAHE (*Siddhartha & Santra, 2020*), intensity normalization followed by CLAHE (N-CLAHE) (*Horry et al., 2020*; *El Asnaoui & Chawki, 2020*), Perona-Malik filter (PMF), unsharp masking (UM) (*Rezaul Karim et al., 2020*), Bi-histogram equalization with adaptive sigmoid function (BEASF) (*Haghanifar et al., 2020*), the gamma correction (GC) (*Rahman et al., 2021b*), histogram stretching (HS) (*Wang et al., 2021*; *Zhang et al., 2021*), Moment Exchange algorithm (MoEx), CLAHE (*Lv et al., 2021*), local phase enhancement (LPE) (*Qi et al., 2021*), image contrast enhancement algorithm (ICEA) (*Canayaz, 2021*), and Gaussian filter (*Medhi, Jamil & Hussain, 2020*) are, as far as we are aware, the only adopted techniques in COVID-19 recognition to date. An overview of these works is listed in Table 1. It should be noted that the CLAHE algorithm has widely used by the majority, while some pursued a hybridization method. Moreover, the utilized filters can result in blurry (by Gaussian filter) or blocky (by PMF) features in the processed image. Accordingly, there is still room to incorporate more effective preprocessing techniques to further increase the accuracy of these systems.

Motivated by the outstanding results in the previously mentioned works as well as the need for close-to-perfect recognition models, this paper integrates novel image preprocessing enhancement with deep learning to meet the challenges arising from data deficiency and complexity. Specifically, we combine an adaptive median filter (AMF) and a non-local means filter (NLMF) to remove the noise from the images. Numerous works have already analyzed the performance of these two filters for denoising x-ray imagery (*Kim, Choi & Lee, 2020*; *Raj & Venkateswarlu, 2012*; *Rabbouch, Messaoud & Saâdaoui, 2020*; *Sawant et al., 1999*; *Mirzabagheri, 2017*), demonstrating their superiority over various filters, including the ones in the cited works in terms of removing impulsive, Poisson, and speckle noise while preserving the useful image details. We then utilize the CLAHE approach that has been already applied for the enhancement of contrast in medical images (*Zhou et al., 2016*; *Sonali et al., 2019*; *Wen, Qi & Shuang, 2016*), to enhance the contrast of the denoised images. The enhanced images are finally fed into the state-of-the-art convolution neural network (CNN) called MobileNet (*Howard*

**Table 1 An overview of image enhancement techniques and the deep learning method used for COVID-19 detection.**

| Study | Image enhancement appraoch | Method |
|---|---|---|
| *Civit-Masot et al. (2020)* | HE | VGG16 |
| *Tartaglione et al. (2020)* | HE | ResNet18, ResNet50, DenseNet121 |
| *Ramadhan et al. (2020)* | CLAHE | COVIDLite |
| *El-bana, Al-Kabbany & Sharkas (2020)* | CLAHE | InceptionV3 |
| *Saiz & Barandiaran (2020)* | CLAHE | VGG16 |
| *Maguolo & Nanni (2021)* | CLAHE | AlexNet |
| *Punn & Agarwal (2021)* | ATV | ResNet, InceptionV3, InceptionResNetV2, DenseNet169, and NASNetLarge |
| *Siddhartha & Santra (2020)* | White balance, CLAHE | COVIDLite |
| *Horry et al. (2020)* | N-CLAHE | VGG19 |
| *El Asnaoui & Chawki (2020)* | CLAHE | VGG16, VGG19, DenseNet201, InceptionResNetV2, InceptionV3, Resnet50, and MobileNetV2 |
| *Rezaul Karim et al. (2020)* | HE, PMF, UM | DeepCOVIDExplainer |
| *Medhi, Jamil & Hussain (2020)* | Gaussian filtering | Deep CNN |
| *Haghanifar et al. (2020)* | CLAHE, BEASF | COVID-CXNet (UNet+DenseNet) |
| *Rahman et al. (2021b)* | GC | Seven different deep CNN networks for classification and modified Unet network for segmentation |
| *Wang et al. (2021)* | HS | PatchShuffle Stochastic Pooling NN |
| *Zhang et al. (2021)* | HS | Deep convolutional attention network |
| *Lv et al. (2021)* | MoEx, CLAHE | Cascade-SEME net |
| *Qi et al. (2021)* | LPE | Fus-ResNet50 |
| *Canayaz (2021)* | ICEA | MH-COVIDNet |

*et al., 2017*), which has been recently utilized for the same classification task by (*Apostolopoulos, Aznaouridis & Tzani, 2020*; *Apostolopoulos & Mpesiana, 2020*). MobileNets are small, low-latency, low-power models parameterized to meet the resource constraints of a variety of use cases. The motivation behind choosing a MobileNet CNN is that it not only helps to reduce overfitting but also runs faster than a regular CNN and has significantly fewer parameters (4.24) (*Howard et al., 2017*; *Yu et al., 2020*). Moreover, MobileNets employ two global hyperparameters based on depthwise separable convolutions to strike a balance between efficiency and accuracy.

KL divergence is one of the measures that reflect the distribution divergence between different probabilities, which has been widely used in the problem of classification imbalanced datasets (*Su et al., 2015*; *Feng et al., 2018*). The KL divergence loss function is more commonly used when using models that learn to approximate a more complex function than simply multiclass classification, such as in the case of an autoencoder used for learning a dense feature representation under a model that must reconstruct the original input. Indeed, the lack of necessary extracted features from the images sometimes cannot provide expected accuracy in the classification result. In this work, inspired by the variational autoencoder learning (*Kingma & Welling, 2013*; *Alfasly et al., 2019*; *Alghaili, Li & Ali, 2020*) the Kullback–Leibler (KL) divergence is adopted to devise more

efficient and accurate representations and measure how far we are from the optimal solution during the iterations. We evaluated the performance of the proposed framework on the COVIDx dataset in terms of a wide variety of metrics: accuracy, sensitivity, specificity, precision, area under the curve, and computational efficiency. Simulation results reveal that the proposed framework significantly outperforms state-of-the-art models from both quantitative and qualitative perspectives.

The novelty of this study is not only to clarify significant features in the CXR images by developing a hybrid algorithm but also proposes a novel approach in how to devise more efficient and accurate by using KL loss. The intent behind this study is not only to achieve a high classification accuracy but to achieve this by training an automated end-to-end deep learning framework based on CNN. This method is superior to transfer learning for evaluating the importance of features derived from imagery, as it is not relying on features previously learned by the pretrained model, which was first trained on nonmedical images. The main contributions of this work can be summarized as follows:

- For COVID-19 recognition, we propose an automated end-to-end deep learning framework based on MobileNet CNN with KL divergence loss function.
- We propose an impressive approach to ensure a sufficiently diverse representation by predicting the output of the mean $\mu$ and standard-deviation $\sigma$ of the Gaussian distribution.
- We incorporate a novel preprocessing enhancement technique consisting of AMF, NLMF, and CLAHE to meet the challenges arising from data deficiency and complexity.
- We analyze the performance of the preprocessing enhancement scheme to demonstrate its role in enhancing the discrimination capability of the proposed model.

The rest of this paper is organized as follows: "Proposed Method" describes the phases of the proposed method. "Results" highlights the experimental results. "Discussion" discusses these results, and the conclusion is presented in the "Conclusion".

## PROPOSED METHOD

In this section, we briefly describe the scenario of the methodology used to achieve the purpose of this study. The proposed method is depicted in Fig 2, which generally consists of two phases: (a) image preprocessing, to overcome the existing drawbacks mentioned in the previous section; (b) training and testing dedicated to image classification.

### Data acquisition

In this work, we used the COVIDx dataset used by *Wang, Lin & Wong (2020)* to train and evaluate the proposed model. In brief, the COVIDx dataset is an open-source dataset that can be downloaded from https://github.com/lindawangg/COVID-Net/blob/master/docs/COVIDx.md. The instructions given by *Wang, Lin & Wong (2020)* were followed to set up the new dataset. Since few CXR images of positive COVID-19 cases are available, we downloaded more COVID-19 x-ray images from https://github.com/ml-workgroup/covid-19-image-repository, and from https://github.com/armiro/COVID-CXNet/tree/master/chest_xray_images/covid19. Duplicated images were omitted from the new dataset

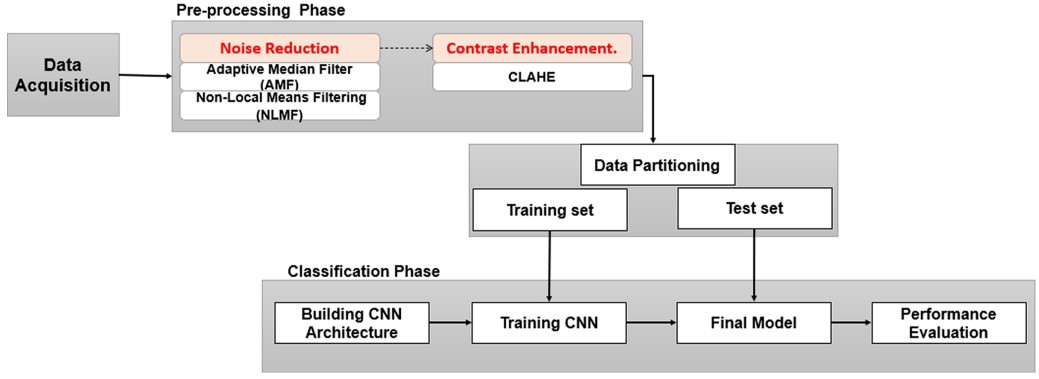

**Figure 2 Framework of study.**               

**Table 2 The number of images for each class.**

| Classes | Total | Training set 70% | Validation set 30% | Test set (unseen) |
|---|---|---|---|---|
| COVID-19 | 2,128 | 1,420 | 608 | 100 |
| Normal | 8,066 | 5,027 | 2,154 | 885 |
| Pneumonia | 5,575 | 3,487 | 1,494 | 594 |
| Total | 15,769 | 9,933 | 4,257 | 1,579 |

to ensure that the proposed training model is more accurate. Thus, the actual number of images in the COVID-19 class is 2,128 instead of the 1,770 images from COVIDx (updated on January 28, 2021). We used the same test set that was used for evaluation by *Wang, Lin & Wong (2020)*, making only a slight change by increasing the number of COVID-19 images to 100 instead of 92. We further split the training data keeping 70% data for training and 30% data for validation. Table 2 summarizes the number of images in each class and the total number of images used for training and testing.

## Data preprocessing method

In this study, we attempt to provide an algorithm that would increase the image quality by using a hybrid technique consisting of noise reduction and contrast enhancement. Specifically, two efficient filters are used for noise reduction while CLAHE is used for contrast enhancement. The first filter is the AMF, which removes impulse noise (*Ning, Liu & Qu, 2009*; *Khare & Chugh, 2014*). This filter is followed by the NLMF algorithm that calculates similarity based on patches instead of pixels. Given a discrete noisy image $u = u(i)$ for pixel $I$, the estimated value of $NL[u](i)$ is the weighted average of all pixels:

$$NL[u](i) = \sum_{j \in i} w(i,j).u(j), \tag{1}$$

where the weight family $w(i,j)j$ depends on the similarity between the pixels $i$ and $j$.

The similarity between the two pixels $i$ and $j$ is defined by the similarity of the intensity of gray-level vectors $u(N_i)$ and $u(N_j)$, where $N_l$ signifies a square neighborhood of fixed size and centered at a pixel $L$. The similarity is measured as a function to minimize the

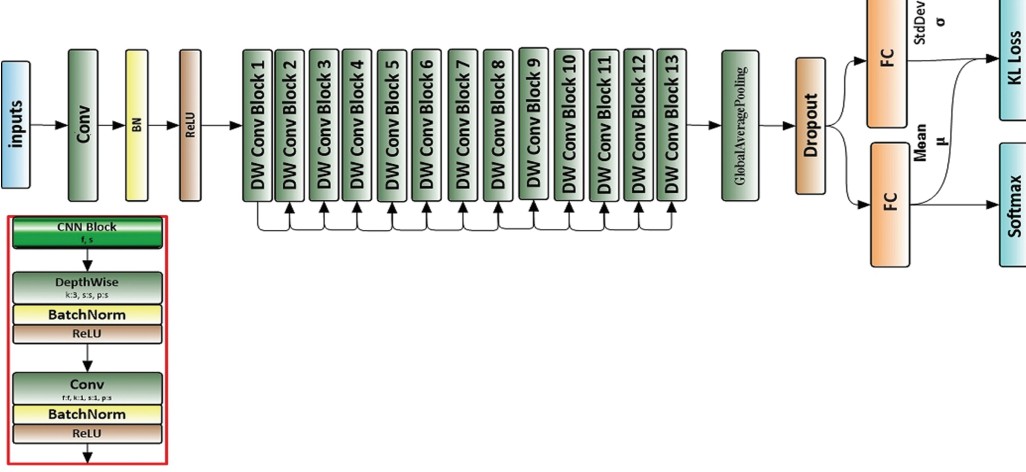

**Figure 3 Architecture of proposed neural network.**

weighted Euclidean distance, $\|u(N_i) - u(N_j)\|^2_{(2,a)}$ where $a > 0$ is the Gaussian kernel standard deviation. The pixels with a similar gray-level neighborhood with $u(N_i)$ have larger weights in average. These weights are defined as;

$$w(i,j) = \frac{1}{Z(i)} e^{-\frac{\|u(N_i) - u(N_j)\|^2_{(2,a)}}{h^2}}, \tag{2}$$

where $Z(i)$ is the normalizing constant: $Z(i) = \sum_j e^{-\frac{\|u(N_i) - u(N_j)\|^2_{(2,a)}}{h^2}}$, and the parameter $h$ acts as a degree of filtering.

Next, CLAHE is applied to the denoised images to achieve an acceptable visualization and to compensate for the effect of filtration that may contribute to some blurring on the images (*Huang et al., 2016*; *Senthilkumar & Senthilmurugan, 2014*). Since there are many homogeneous regions in medical images, CLAHE is suitable for optimizing medical images as the CLAHE algorithm creates non-overlapping homogeneous regions.

## Classification neural network model

We used a deep neural network structure called a MobileNet neural network (*Howard et al., 2017*). All images were resized to 224 × 224 × 3 before being used as input to the neural network. Figure 3 depicts the architecture of the proposed neural network.

Apart from the first layer, which is a full convolution, the MobileNets are constructed using depthwise separable convolutions. Depthwise separable convolution is a factorized convolution that factorizes the standard convolution into a depthwise convolution and a 1 × 1 convolution called pointwise convolution. This procedure reduces the computations and model size drastically. The overall architecture of the MobileNet is shown in Table 3.

The deep convolutional neural network is used to extract high context features per input instance. The global average pooling layer is used here to reduce the spatial dimensions of the features extracted. The output is a feature vector of size 1,024 for each time step. Then, a dropout layer is used with a probability of 0.001. The output of the

**Table 3 Layers of prposed CNN model architecture.182.**

| Type | Stride | Filter shape | Size in | Size out |
|---|---|---|---|---|
| Conv1 | 2 | $3 \times 3 \times 3 \times 32$ | $224 \times 224 \times 3$ | $112 \times 112 \times 32$ |
| Conv2 dw | 1 | $3 \times 3 \times 32$ | $112 \times 112 \times 32$ | $112 \times 112 \times 32$ |
| Conv2 pw | 1 | $1 \times 1 \times 32 \times 64$ | $112 \times 112 \times 32$ | $112 \times 112 \times 64$ |
| Conv3 dw | 2 | $3 \times 3 \times 64$ | $112 \times 112 \times 64$ | $56 \times 56 \times 64$ |
| Conv3 pw | 1 | $1 \times 1 \times 64 \times 128$ | $56 \times 56 \times 64$ | $56 \times 56 \times 128$ |
| Conv4 dw | 1 | $3 \times 3 \times 128$ | $56 \times 56 \times 128$ | $56 \times 56 \times 128$ |
| Conv4 pw | 1 | $1 \times 1 \times 128 \times 128$ | $56 \times 56 \times 128$ | $56 \times 56 \times 128$ |
| Conv5 dw | 2 | $3 \times 3 \times 128$ | $56 \times 56 \times 128$ | $56 \times 56 \times 128$ |
| Conv5 pw | 1 | $1 \times 1 \times 128 \times 256$ | $28 \times 28 \times 128$ | $28 \times 28 \times 128$ |
| Conv6 dw | 1 | $3 \times 3 \times 256$ | $28 \times 28 \times 256$ | $28 \times 28 \times 265$ |
| Conv6 pw | 1 | $1 \times 1 \times 256 \times 256$ | $28 \times 28 \times 256$ | $28 \times 28 \times 256$ |
| Conv7 dw | 2 | $3 \times 3 \times 256$ | $28 \times 28 \times 256$ | $14 \times 14 \times 256$ |
| Conv7 pw | 1 | $1 \times 1 \times 256 \times 512$ | $14 \times 14 \times 256$ | $14 \times 14 \times 512$ |
| Conv8-12 dw | 1 | $3 \times 3 \times 512$ | $14 \times 14 \times 512$ | $14 \times 14 \times 512$ |
| Conv8-12 pw | 1 | $1 \times 1 \times 512 \times 512$ | $14 \times 14 \times 512$ | $14 \times 14 \times 512$ |
| Conv13 dw | 2 | $3 \times 3 \times 512$ | $14 \times 14 \times 512$ | $7 \times 7 \times 512$ |
| Conv13 pw | 1 | $1 \times 1 \times 512 \times 1,024$ | $7 \times 7 \times 512$ | $7 \times 7 \times 1,024$ |
| Conv14 dw | 2 | $3 \times 3 \times 1,024$ | $7 \times 7 \times 1,024$ | $7 \times 7 \times 1,024$ |
| Conv14 pw | 1 | $1 \times 1 \times 1,024 \times 1,024$ | $7 \times 7 \times 1,024$ | $7 \times 7 \times 1,024$ |
| GAP | 1 | Pool $7 \times 7$ | $7 \times 7 \times 1,024$ | $1 \times 1 \times 1,024$ |
| Dropout | 1 | Probability = 0.001 | $1 \times 1 \times 1,024$ | $1 \times 1 \times 1,024$ |
| FC ($\mu$) | 1 | $128 \times 3$ | $1 \times 1 \times 1,024$ | $1 \times 1 \times 128$ |
| FC ($\sigma$) | 1 | $128 \times 3$ | $1 \times 1 \times 1,024$ | $1 \times 1 \times 128$ |
| Softmax | 1 | Classifier | $1 \times 1 \times 128$ | $1 \times 1 \times 3$ |

dropout layer goes to two fully connected layers that generate an output of size 128. One fully connected layer is used to predict the mean $\mu$, which is used to extract the most significant features from those features extracted in previous layers. The other is used to predict the standard deviation $\sigma$ of a Gaussian distribution, which is used to calculate the KL loss function. The output of the fully connected layer, which used to predict the mean $\mu$ goes to the last layer (Softmax classifier), which is defined by

$$L_{CE}(o, v) = - \sum_{i=1}^{v} o_i \log \left( \frac{e^{pi}}{\sum_{j}^{v} e^{pj}} \right),$$

(3)

where $v$ indicates the output vector, $o$ indicates the objective vector, and $pj$ indicates the input to the neuron $j$.

The categorical cross-entropy loss function is generally used to address such a multiclass classification problem. The three classes are provided with labels such as "0" being a COVID-19 case, "1" being a normal case, and "2" being pneumonia. We adopted Kullback–Leibler divergence loss function to devise more efficient and accurate representations. Moreover, the combined KL loss with the categorical cross-entropy loss

function would enforce the network to give a consistent output, in addition to the preprocessing applied to the input image. The KL divergence distribution between the $\mu;\sigma$ and the prior is considered as a regularization that aids in addressing the issue of overfitting. KL loss function is defined by

$$D_{KL} = -\frac{1}{2}\sum_{i=1}^{n}\left(1 + \log(\sigma_i) - \mu_i^2 - \sigma_i\right), \tag{4}$$

where $n$ is the output vector of the average pooling layer with the size of 1,024, $\mu$ is the mean, which is predicted from one fully connected layer, and $\sigma$ is the standard deviation of a Gaussian distribution, which is predicted from the other fully connected layer in the network, Fig. 3. The multitask learning loss function for our proposed network is now defined by

$$L = \alpha D_{KL} + L_{CE}(o, v), \tag{5}$$

We use a weighted loss function as illustrated in Eq. (5). The weight of KL loss $\alpha$ is empirically set to (0:1) to be used as a one-hot vector, which not only ensures a clear representation of the true class, but also helps in addressing the large variance arising due to unbalanced data.

## Experiments

All CXRs were resized to the same dimension of 224 × 224 in .jpg format. In the first phase, the AMF window size was taken to be 5 × 5 for effective filtering. The resultant image was then subjected to the NLMF technique. The performance of the NLMF was depended on 7 × 7 of the search window, 5 × 5 of the similarity window, and a degree of filtering $h = 1$. Furthermore, we increased the contrast using CLAHE with the bin of 256 and block size of 128 in slope 3 to get the enhanced images. We passed the images to KL-MOB as the input to predict the CXR image (COVID-19, normal, or pneumonia). Because many functions are not built-in functions from deep learning libraries, such as the relu6 activation function with a max value of six, we built an interface for the evaluation process that contains all layers in the network, as in a training network, but which is not used for training. Instead, it is used to pass on the input image to produce the output.

The proposed model (KL-MOB) is implemented by using the Python programming language. All experiments were conducted on a Tesla K80 GPU graphics card on Google Collaboratory with an Intel© i7-core @3.6GHz processor and 16GB RAM with 64-bit Windows 10 operating system. The original and enhanced images are used separately to train the KL-MOB. In the first stage, the baseline model is trained to verify the influence of the KL loss on performance. Figure 4 presents the curve comparisons of all training processes. With the maximum training epoch set to 200. A large gap between training and validation in both original and enhanced images indicates the presence of overfitting.

The network is trained by using a SoftMax classifier with an Adam optimizer (*Kingma & Ba, 2014*) with the initial learning rate set to 0.0001 and a batch size of 32.

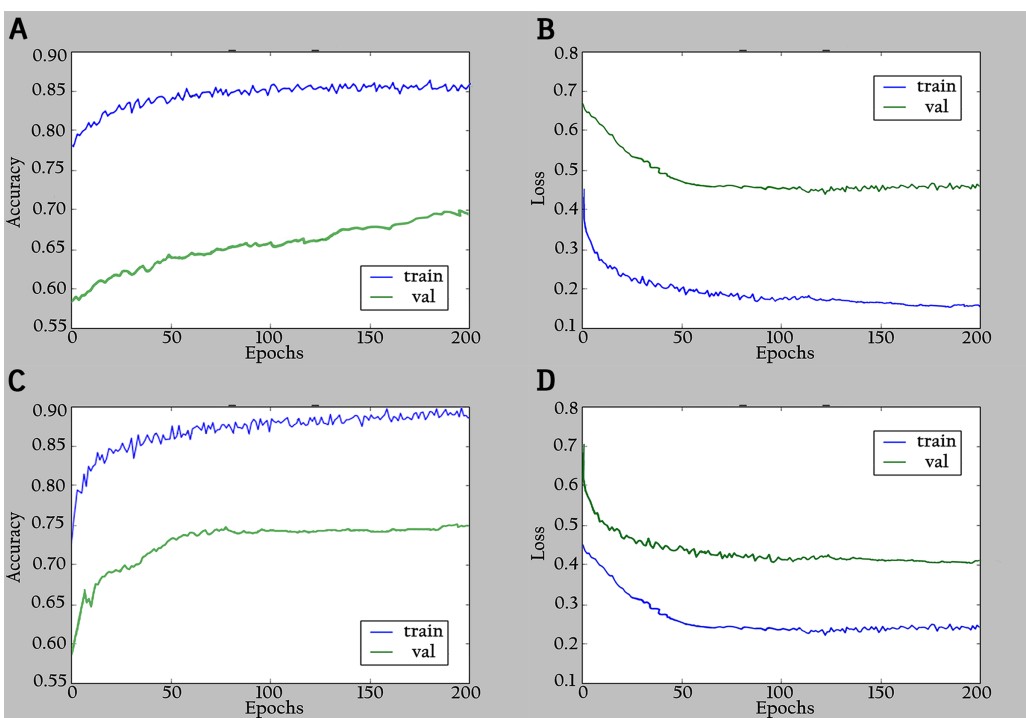

**Figure 4** Accuracy and loss graphs for baseline model: (A) training and validation accuracy of the original images, (B) training and validation loss of the original images, (C) training and validation accuracy of the enhanced images and (D) training and validation loss of the enhanced images.

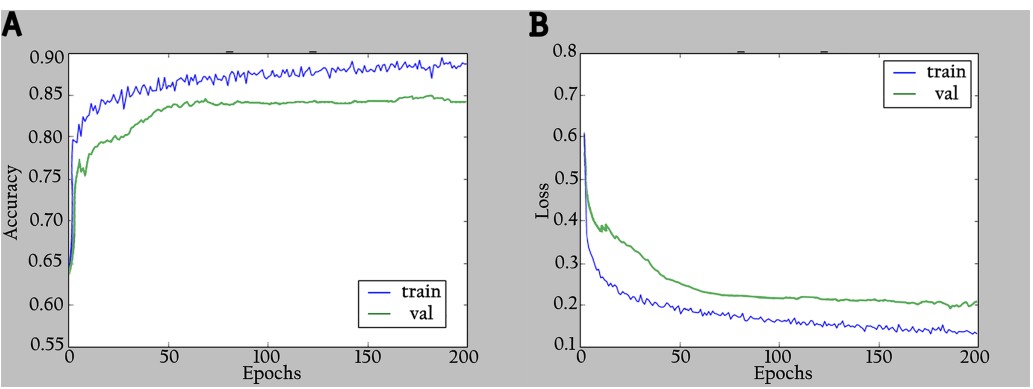

**Figure 5** Accuracy and loss graphs for KL-MOB on training and validation of the original images: (A) accuracy and (B) loss.

The dataset used for training is divided into 70% as a training set and 30% as a validation set. The total number of parameters is 3,488,426, where the number of trainable parameters is 3,466,660, and the nontrainable parameters are 21,766. In the training period, 200 epochs were completed to check the KL-MOB model accuracy and loss, which are shown in Figs. 5 and 6.

Beforehand, the impact of different feature sizes on training accuracy has been investigated *via* conducting extensive experiments. Original images perform best when the

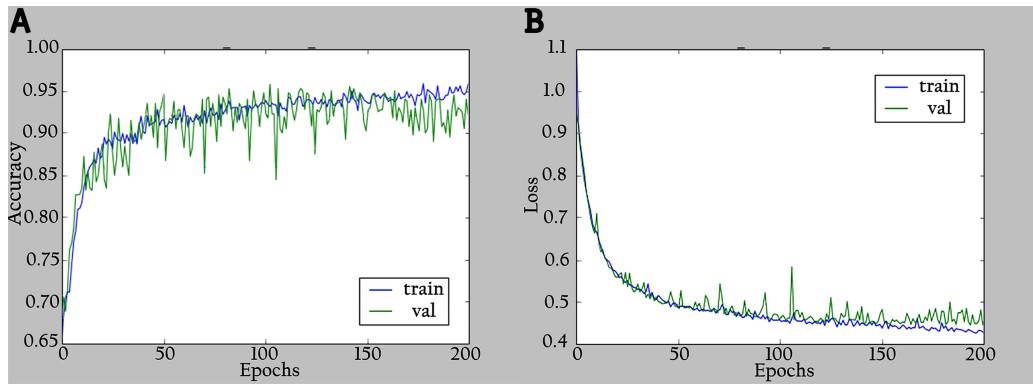

**Figure 6 Accuracy and loss graphs for KL-MOB on training and validation of the enhanced images:**
**(A) accuracy and (B) loss.**

**Table 4 Model performance on different feature sizes.**

| Model | Output vector | Accuracy% | |
|---|---|---|---|
| | | Enhanced | Original |
| | 64 | 93.26 | 88.31 |
| | 128 | 96.06 | 89.36 |
| KL-MOB | 256 | 95.87 | 93.24 |
| | 512 | 94.83 | 91.08 |
| | 1024 | 94.47 | 90.38 |

length is set to 256, with an accuracy of 93.24%, whereas the enhanced images perform best when the length is set to 128 with an accuracy of 96.06%, as shown in Table 4. This can be attributed to the fact that the KL divergence between $\mu$; $\sigma$ distribution and the prior is considered as a regularization which helps to overcome the overfitting problem.

## Performance evaluation

### Preprocessing performance evaluation

The performance of the proposed preprocessing technique was quantified by using various evaluation metrics such as mean average error (MAE) and peak signal-to-noise Ratio (PSNR). These metrics are desirable because they can be rapidly quantified.

Definition: $x(i,j)$ denotes the samples of the original image, $y(i, j)$ denotes the samples of the output image. $M$ and $N$ are the number of pixels in row and column directions, respectively. *MAE* is calculated as in Eq. (6), where a large value means that the images are of poor quality.

$$MAE = |E(x) - E(y)|, \qquad (6)$$

The limited value *PSNR* implies that the images are of low quality. PSNR is described in terms of Mean Square Error *MSE* as follows:

$$PSNR = 10 \log_{10} \frac{MAX_i^2}{MSE}, \qquad (7)$$

**Table 5 Average PSNR (db) and MAE for the various noise-reduction methods.**

| Method | Covid19 | | Normal | | Pneumonia | |
|---|---|---|---|---|---|---|
| | PSNR | MAE | PSNR | MAE | PSNR | MAE |
| AMF | 21.91 | 14.46 | 21.19 | 17.88 | 20.43 | 19.47 |
| NLMF | 20.47 | 19.19 | 20.41 | 19.41 | 20.40 | 19.40 |
| Proposed method | 22.04 | 14.38 | 21.21 | 17.59 | 20.45 | 19.32 |

where $MAX_i^2$ is the maximum possible pixel intensity value 255 when the pixel is represented by 8 bits.

$$MSE = \sqrt{\frac{1}{MN} \sum_{i=1}^{M-1} \sum_{j=1}^{N-1} [x(i,j) - y(i,j)]^2} \qquad (8)$$

,

*Neural network performance evaluation*

The test set described in the previous section was used to evaluate KL-MOB. The classification outcome has four cases: True Positive (TP), False Positive (FP), True Negative (TN) and False Negative (FN). The metrics used to measure the performance are accuracy (ACC), sensitivity (TPR), specificity (SPC), and precision (PPV) and are defined as follows:

$$Accuracy\ (ACC) = \frac{TP + TN}{TP + FP + TN + FN}, \qquad (9)$$

$$Sensitivity\ (TPR) = \frac{TP}{TP + FN}, \qquad (10)$$

$$Specificity\ (SPC) = \frac{TN}{FP + TN}, \qquad (11)$$

$$Precision\ (PPV) = \frac{TP}{TP + FP}, \qquad (12)$$

The graph of true positive rate (TPR) and false positive rate (FPR) is the receiver operating characteristic (ROC) curve. The FPR is calculated as follows:

$$False\ Positive\ Rate\ (FPR) = \frac{FP}{FP + TN}. \qquad (13)$$

## RESULTS

In the experiments, noise reduction and contrast enhancement performance were evaluated independently, since they are two separate issues. The average value was computed for all images in each class. Tables 5 and 6 show the results for noise reduction and image enhancement, respectively. Figure 7 shows the noise reduction techniques that were applied to the original image and the hybrid method used in this work. Although the denoising filters could smooth and blur the resulting images, this can be enhanced by improving the image edges and by highlighting the high-frequency components to

**Table 6 Average PSNR (db) and MAE for the various contrast-enhancement methods.**

| Method | Covid19 | | Normal | | Pneumonia | |
|---|---|---|---|---|---|---|
| | PSNR | MAE | PSNE | MAE | PSNR | MAE |
| CLAHE | 17.83 | 27.35 | 17.12 | 25.98 | 21.91 | 16.20 |
| Proposed method | 19.14 | 23.13 | 17.28 | 25.45 | 22.11 | 16.01 |

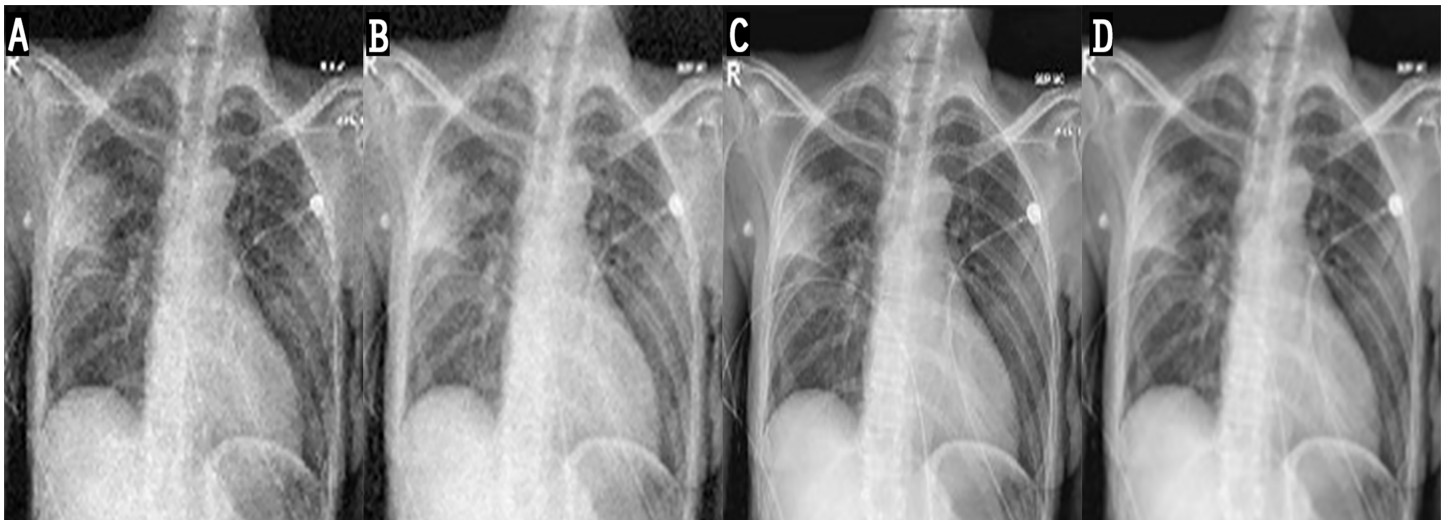

**Figure 7 Result of noise-reduction techniques applied to images: (A) original image, (B) imagedenoised by AMF, (C) image denoised by NLMF, (D) image denoised by proposed method.**

remove the residual noise. Figure 8 displays the original images and their enhanced versions.

The performance of the proposed KL-MOB was evaluated separately for each class of the test set. Table 7 compares the performance of the KL-MOB model for the classification problem involving original and enhanced images. Note that the proposed method boosts the performance of the KL-MOB model in COVID-19 detection, as shown in Figs. 9 and 10.

## DISCUSSION

This work proposes an approach that combines noise-reduction algorithms with contrast enhancement. This approach introduces a type of hybrid filtering and contrast enhancement for the data set of images used for COVID-19 detection. The well-known measurable methods PSNR and MAE were used as image quality measurements for assessing and comparing image quality. The results of Table 5 show that using an AMF followed by a NLMF is entirely favorable for eliminating noise. The proposed hybrid algorithm is applied to the entire image instead of just parts of the image and preserves important details. Figure 11 illustrates the difference between the original CXRs and CXRs enhanced by applying the method proposed herein. Furthermore, we judge the lung

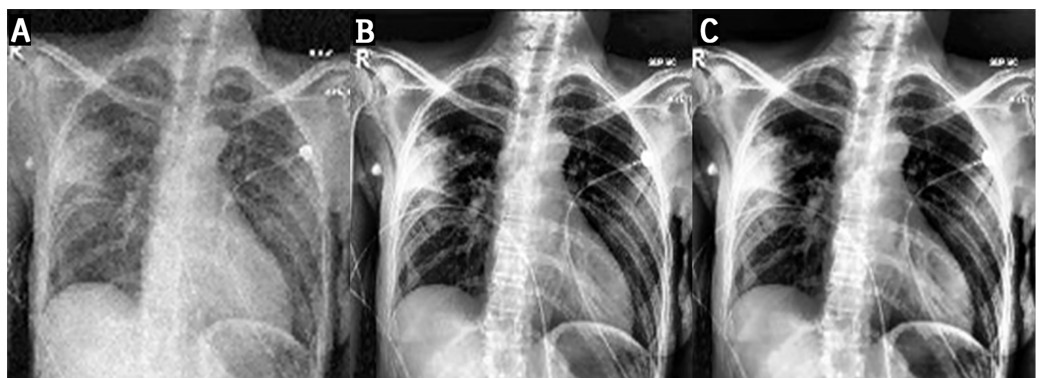

**Figure 8 Results of image enhancement: (A) original image, (B) image enhanced by CLAHE, (C) image enhanced by proposed method.**

**Table 7 Metrics for original images and for images enhanced by KL-MOB.**

|  | Enahnced image | | | | | Original image | | | | |
|---|---|---|---|---|---|---|---|---|---|---|
|  | ACC% | PPV% | SPC% | TPR% | MCC% | ACC% | PPV% | SPC% | TPR% | MCC% |
| Covid19 | 99.87 | 99.00 | 99.93 | 99.00 | 98.93 | 92.61 | 96.83 | 99.13 | 74.39 | 80.60 |
| Normal | 98.24 | 98.30 | 97.85 | 98.64 | 96.53 | 97.11 | 98.17 | 98.99 | 93.86 | 93.77 |
| Pneumonia | 97.99 | 97.81 | 98.68 | 97.31 | 96.03 | 91.00 | 81.30 | 86.74 | 98.26 | 82.53 |
| Overall | 98.70 | 98.37 | 98.82 | 98.32 | 96.60 | 93.57 | 92.10 | 94.95 | 88.84 | 85.90 |

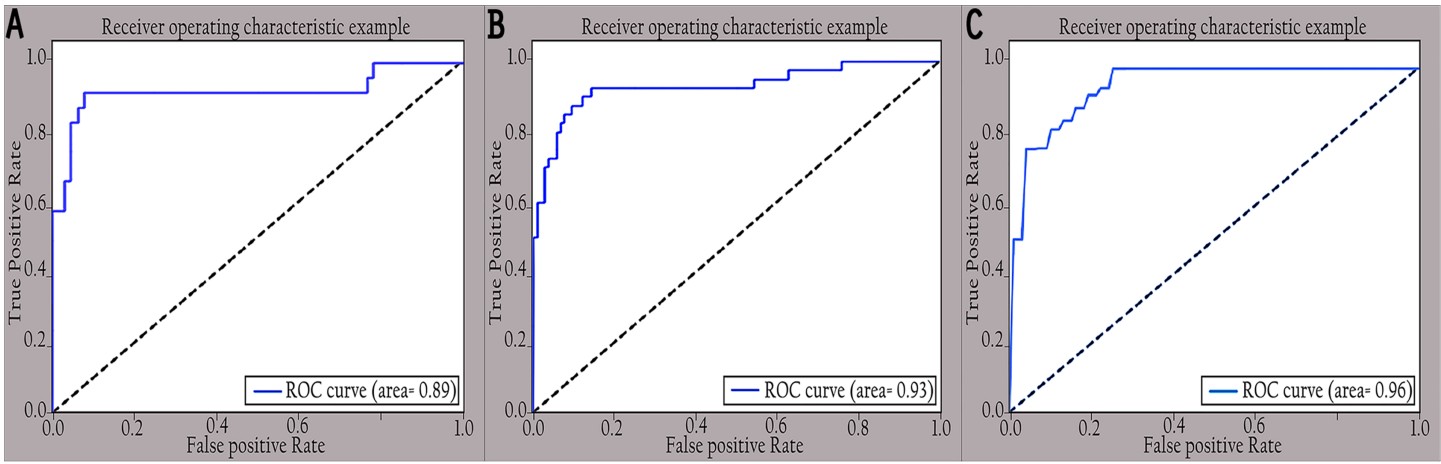

**Figure 9 ROC curves of different classes for original images: (A) COVID-19, (B) normal and (C) pneumonia.**

damage in the enhanced image to be more perspicuous than in the original image. In addition, CLAHE with a bin of 256 gives the best PSNR, as shown in Table 6.

To show the impact of the KL divergence loss on the efficacy of the proposed method, we performed several experiments using the categorical entropy loss function (CCE) and

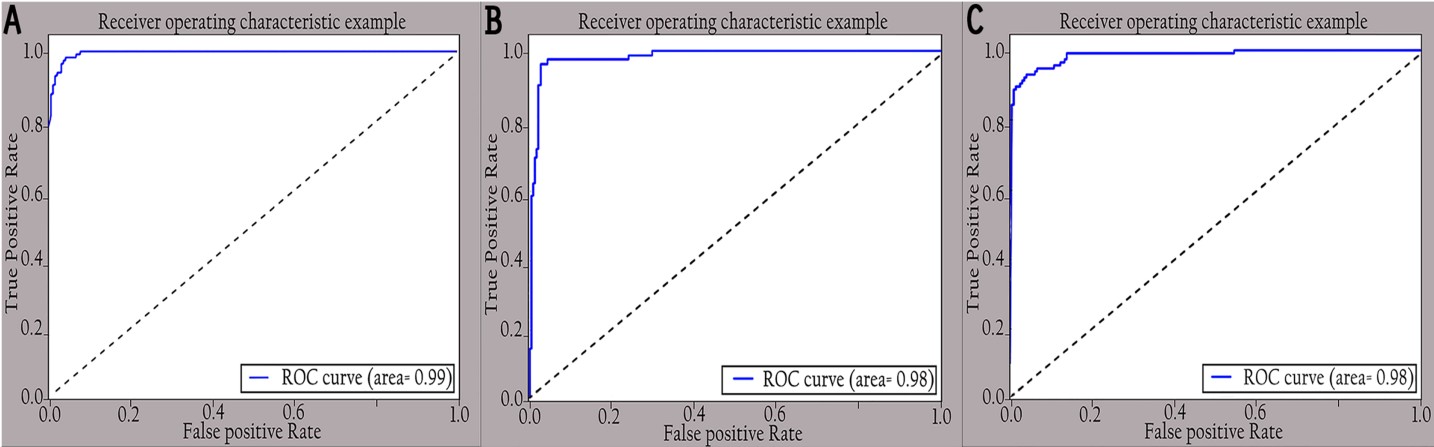

**Figure 10** ROC curves of different classes for enhanced images: (A) COVID-19, (B) normal and (C) pneumonia.

**Table 8 Performance on the test set with different loss functions.**

| Model | Loss function | Enhanced image | | | | Original image | | | |
|-------|---------------|------|------|------|------|------|------|------|------|
| | | ACC% | PPV% | SPC% | TPR% | ACC% | PPV% | SPC% | TPR% |
| KL-MOB | CCE | 96.79 | 95.22 | 97.60 | 95.42 | 90.14 | 87.94 | 92.23 | 83.05 |
| | MSE | 92.50 | 89.70 | 94.16 | 86.92 | 85.12 | 94.53 | 97.50 | 95.11 |
| | Proposed method | 98.70 | 98.37 | 98.82 | 98.32 | 93.57 | 92.10 | 94.95 | 88.84 |

the mean square error (MSE) loss function. The results obtained in Table 8 show that the proposed method has a great impact on the performance of KL-MOB, thereby justifying the selection of the proposed network architecture and its associated training/learning schemes.

Figure 12 shows the confusion matrix of the proposed network: all classes are identified with high true positives. Note that the COVID-19 cases are 99% correctly classified by the KL-MOB model. Only 1% of COVID-19 cases are misclassified as pneumonia (non-COVID-19), and 1.4% of the normal cases are misclassified as pneumonia. Only 0.2% of pneumonia (non-Covid-19) cases are wrongly classified as COVID-19. These results demonstrate that the proposed KL-MOB has a strong potential for detecting COVID-19. In particular, with limited COVID-19 cases, the results show that no confusion arises between normal patients and COVID-19 patients.

In our experiment of 100 patients with COVID-19, only one was misclassified with a 99.0% PPV for COVID-19, which compares favorably with previous results of 98.9% and 96.12% for *Wang, Lin & Wong (2020)* and *Rezaul Karim et al. (2020)*, respectively. In addition, we compare the results obtained from the KL-MOB model with those from previous studies that used the same or similar datasets for evaluation (see Table 9). Not included in the comparison are studies that used smaller datasets (*Farooq & Hafeez, 2020*; *Afshar et al., 2020*; *Hirano, Koga & Takemoto, 2020*; *Ucar & Korkmaz, 2020*). The results

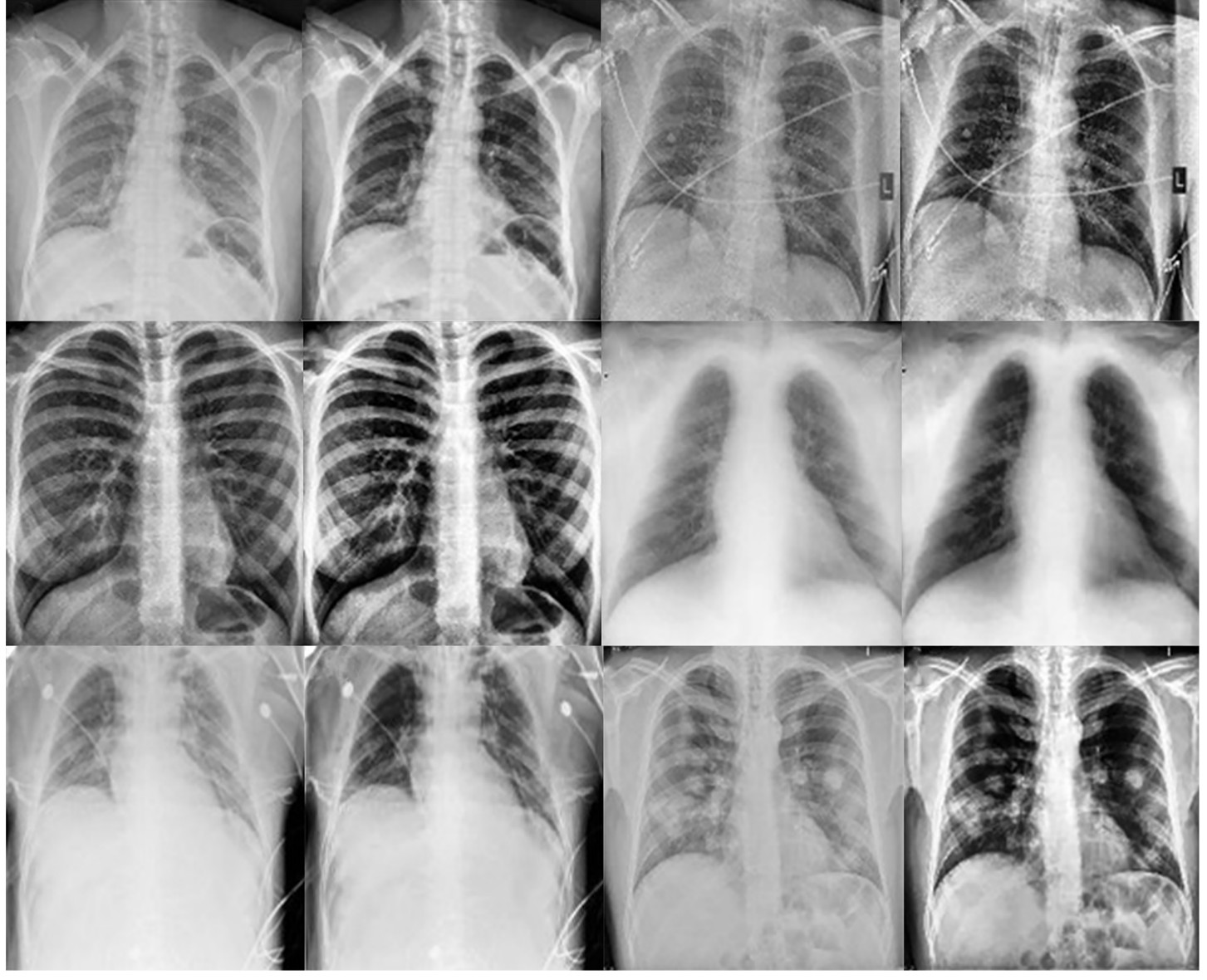

**Figure 11** **The first and third columns show the original images, and the second and fourth columnsshow the corresponding enhanced images.**

show that, for all performance metrics [accuracy, sensitivity (TPR), specificity, and PPV for overall detection], the KL- MOB model produces superior results compared with the models of *Wang, Lin & Wong (2020)* and *Rezaul Karim et al. (2020)*.

The promising deep learning models used for the detection of COVID-19 from radiography images indicate that deep learning likely still has untapped potential and can play a more significant role in fighting this pandemic. There is definitely still room for improvement through: (a) the other preprocesses such as increasing the number of images, implementing another preprocessing technique, *i.e.*, data augmentation, utilizing different noise filters, and enhancement techniques. (b) design a model that deals with multiple

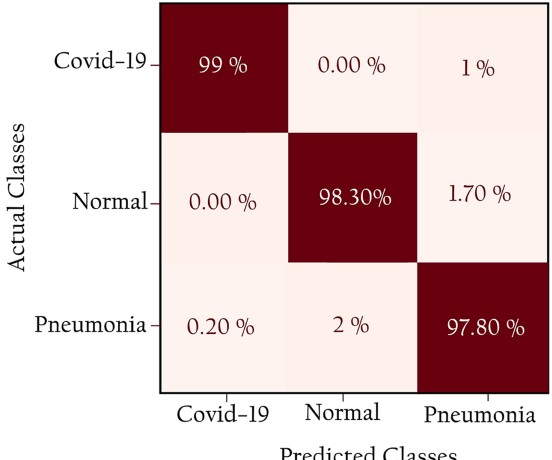

**Figure 12 Confusion matrix for KL-MOB applied to COVIDx test dataset.**

**Table 9 Comparative performance of the various models with the improvement percentage compared tothe state of art.**

| Study | Classifier | ACC% | SPC% | TPR% | PPV% |
|---|---|---|---|---|---|
| *Wang, Lin & Wong (2020)* | COVID-Net (large) | 95.56 | 96.67 | 93.33 | 93.55 |
| *Ahmed et al. (2020)* | ReCoNet | 97.48 | 97.39 | 97.53 | 96.27 |
| *Rezaul Karim et al. (2020)* | DeepCOVIDExplainer | 98.11 | 98.19 | 95.06 | 96.84 |
| Proposed method | KL-MOB | 98.7 | 98.82 | 98.32 | 98.37 |
| % Improvement | | 0.60 | 0.64 | 3.43 | 1.58 |

inputs simultaneously, where utilizing multiple modalities may achieve superior outcomes than the individual modality (*Zhang et al., 2021*).

## CONCLUSION

This work proposes a novel CNN-based MobileNet-structured neural network for detecting COVID-19 using COVIDx, which is the most widely used public dataset of CXR images to date. The evaluation of this approach shows that it outperforms the recent approach in terms of accuracy, specificity, sensitivity, and precision (98.7%, 98.%, 98.32% and 98.37%, respectively). The proposed method relies on image manipulation by applying a hybrid technique to enhance the visibility of CXR images. This advanced preprocessing technique facilitates the task of the KL-MOB model to extract features, allowing complex patterns in medical images to be recognized at a level comparable to that of an experienced radiologist. The KL divergence is used to boost the performance of the KL-MOB model, which outperforms recent approaches, as shown by the results. The KL divergence between the $\mu;\sigma$ distribution and the prior is considered as a regularization, which aids to overcome the overfitting problem. Moreover, it is also believed that the notion of using KL divergence can be extended to other similar scenarios such as content-based image retrieval and fine-grained classification to improve the quality

of object representation. Considering several essential factors such as the pattern by which COVID-19 infections spread, image acquisition time, scanner availability, and costs, we hope that these findings will make a useful contribution to the fight against COVID-19 and increase the acceptance of artificial-intelligence-assisted applications in clinical practice.

In future work, we will further enhance the proposed method's performance by including lateral views of CXR images in the training data because, in some cases, frontal-view CXR images do not permit a clear diagnosis of pneumonia cases. Besides, this work lacked in applying some of the techniques such as progressive resizing (*Bhatt, Ganatra & Kotecha, 2021a*), which can be applied to CNNs to carry out imaging-based diagnostics. Furthermore, visual ablation studies (*Bhatt, Ganatra & Kotecha, 2021b*; *Joshi, Walambe & Kotecha, 2021*; *Gite et al., 2021*) can be performed along with deep learning, which will significantly improve the detection of COVID-19 manifestations in the CXR images. Since only a limited number of CXR images are available for COVID-19 infection, out-of-distribution issues may arise, so more data from related distributions is needed for further evaluation. There are several techniques that would be another way to overcome this problem, include, but are not limited to data augmentation techniques (*Chaudhari, Agrawal & Kotecha, 2019*), transfer learning (*Taresh et al., 2021*; *Bhatt, Ganatra & Kotecha, 2021a*), domain-adaptation (*Zhang et al., 2020*; *Jin et al., 2021*) and adversarial learning (*Goel et al., 2021*; *Rahman et al., 2021a*; *Motamed, Rogalla & Khalvati, 2021*), etc. Finally, the image enhancement must be verified by a radiologist, which we have not yet been able to do due to the emerging conditions.

### Funding
This work was supported by the National Natural Science Foundation (61572177). No additional external funding received for this study. The funders had no role in study design, data collection and analysis, decision to publish, or preparation of the manuscript.

### Grant Disclosures
The following grant information was disclosed by the authors:
National Natural Science Foundation: 61572177.

### Competing Interests
The authors declare that they have no competing interests.

### Author Contributions
- Mundher Mohammed Taresh conceived and designed the experiments, performed the experiments, analyzed the data, performed the computation work, authored or reviewed drafts of the paper, and approved the final draft.
- Ningbo Zhu performed the computation work, prepared figures and/or tables, and approved the final draft.
- Talal Ahmed Ali Ali conceived and designed the experiments, authored or reviewed drafts of the paper, and approved the final draft.

- Mohammed Alghaili performed the computation work, authored or reviewed drafts of the paper, and approved the final draft.
- Asaad Shakir Hameed analyzed the data, prepared figures and/or tables, and approved the final draft.
- Modhi Lafta Mutar analyzed the data, prepared figures and/or tables, and approved the final draft.

## Data Availability

The data are available at GitHub: -https://github.com/lindawangg/COVID-Net/blob/master/docs/COVIDx.md.

-https://github.com/ml-workgroup/covid-19-image-repository.

-https://github.com/armiro/COVID-CXNet/tree/master/chest_xray_images/covid19.

The python code for training and evaluation are available as Supplemental Files.

## Supplemental Information

Supplemental information for this article can be found online at http://dx.doi.org/10.7717/peerj-cs.694#supplemental-information.

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
