# Peer review of "KL-MOB: automated COVID-19 recognition using a novel approach based on image enhancement and a modified MobileNet CNN"

_PeerJ Computer Science, doi:10.7717/peerj-cs.694_

## Round 0.1 · original submission · Minor Revisions

· Academic Editor

Minor Revisions

The manuscript is well written and the results are good. I ask authors to do corrections suggested by 1st and 3rd reviewer before the manuscript is ready for publication. For establishing novelty as questioned by reviewer 2 I would advise adding one paragraph describing the end-to-end pipeline followed and that will suffice to address the comments by reviewer 2.

·

Basic reporting

Grammatical mistakes need to be rectified.
There are some writing mistakes such as Covid1-9 in line number 145. The manuscript should be thoroughly checked.
Equation-2 should be rewritten to differentiate - sign.
Figure 6 and 7 should be enlarged enough to read values.

Experimental design

Experimental design is OK.
Explanation of proposed architecture needs to be re-write and it should meet academic writing criteria.
How KL-Divergence used and by using it, how your work has been enhanced? Justify.
Authors have added KL-Divergence loss, work should also be compared with other loss methods to show work efficiency.

Validity of the findings

Table 1 shows that classes are imbalanced and large variance. How it is handled during training?
The loss and accuracy graphs need to be presented. Epoch and training explanation needs to present.
Authors can compare his work with baseline Deep CNN classifier models.

Reviewer 2 ·

Basic reporting

The proposed research described the automated COVID-19 recognition using a
novel approach based on image enhancement and a modified MobileNet CNN. The COVID-19 recognition using image processing-related techniques has not remained a novel research area. Secondly, the presented research work does not advance science. So I do not recommend the publication of the manuscript.

Experimental design

The manuscript has numerous organization and research design issues. The authors did not discuss how they collected data, data pre-processing techniques used in the conducted experiments, technical specifications of used equipment, etc. Similarly, the data processing and applications layers lacked sufficient detail for a critical assessment, but my opinion based on the information provided is that these aspects also do not advance the science.

Validity of the findings

The authors have not carried out a rigorous literature review of the existing research works, a detailed comparison analysis, and identification of appropriate research gaps. I recommend that authors review peer-reviewed articles of the last three years of reputed journals that are significantly lacking in the proposed research work. Furthermore, the discussion of the results concerning the proposed research work is not organized correctly. The conclusion section also needs a revision.

Additional comments

The proposed research described the automated COVID-19 recognition using a
novel approach based on image enhancement and a modified MobileNet CNN. The COVID-19 recognition using image processing-related techniques has not remained a novel research area. The presented research work does not advance science even after combining image enhancement and the CNN methodologies. So I do not recommend the publication of the manuscript.

Firstly, the manuscript has numerous organization and research design issues. The authors did not discuss how they collected data, data pre-processing techniques used in the conducted experiments, technical specifications of used equipment, etc. Similarly, the data processing and applications layers lacked sufficient detail for a critical assessment, but my opinion based on the information provided is that these aspects also do not advance the science.

Secondly, the abstract and introduction lack the discussion of the need for the proposed system. I recommend authors carry out a detailed literature survey again, identify open issues, and add an additional section representing the novelty of the proposed approach. I also want to recommend authors carry out some more real-time experiments. Furthermore, the discussion of the results concerning the proposed research work is not organized correctly. The conclusion section also needs a revision.

For all the above reasons, I would not recommend publishing this manuscript. It requires lots of revisions. If authors want to publish this manuscript, then it is essential that they should re-conduct the presented experiments and re-write the manuscript after conducting a rigorous literature survey. After making the above changes, the manuscript may be submitted here or elsewhere.

·

Basic reporting

Clear, unambiguous, technical English language has been used in the paper. Still the manuscript can be improved by avoiding a few grammatical errors.

Sufficient field background/context provided, however I feel Literature review part(preferably in a tabular format) can be enhanced further by adding related papers from the current year i.e. 2021.
A short paragraph on MobileNet can form a baseline for the proposed research so authors are advised to add it in the literature.

Professional article structure, figures, tables. Raw data shared.

Line no. 85 states "The motivation behind choosing Mobile CNN is that it not only helps to reduce overfitting but also runs faster than regular CNN with many fewer parameters (Howard et al., 2017; Yu et al., 2020)".Please specify.


KL divergence loss function must be explained with proper citations and references.

The section "Classification Neural Network Model" line no.127 should be presented in a tabular format to make it readable and understandable.

Experimental design

Original primary research within Aims and Scope of the journal.

Research question well defined, relevant & meaningful.

However, the explicit summary can be included which should state the important aspects of the proposed research to fill an identified knowledge gap.


Methods described with sufficient detail & information to replicate. However there are many Contrast enhancement techniques such as Filtering with morphological operator or Histogram equalization or Median filtering. The notion of choosing CLAHE should be justified with more details.

Training testing percentage not mentioned clearly in Table 1. The number of images for each class.

Line no. 164 needs a full stop.
Line no 173 "Adaptation of such an approach introduced…" reframe this.
Same nomenclature should be used throughout the paper such as table 3 contains "our method" whereas table 5 uses "proposed method"

Validity of the findings

Conclusions are well stated, linked to original research question & limited to supporting results.

% improvement in the evaluation can be added in the results table as the last row to state the impact of the proposed model.


Author should include some small portion of future work mentioning following:
https://www.sciencedirect.com/science/article/pii/S2405844021013141
https://doi.org/10.1155/2021/8828404

What about using Grad CAM for visual explanations of the detection as given in https://peerj.com/articles/cs-348/ and https://ieeexplore.ieee.org/abstract/document/9391727

https://doi.org/10.7717/peerj-cs.340 for Lime tool in XAI

If data is not balanced authors must state some data augmentation techniques and particularly the advanced using GANs as in https://link.springer.com/article/10.1007%2Fs00500-019-04602-2

or in the upcoming domains such as transfer learning/domain adaptation/adversarial learning etc.

Additional comments

Overall a well written paper with impressive results.

---

## Round 0.2 · Minor Revisions

· Academic Editor

Minor Revisions

Please address all the points raised by the reviewer and prepare a new version of the manuscript.

Additionally, please measure all the performances of the binary classification through the Matthews correlation coefficient (MCC) besides the rates already employed.

Reviewer 4 ·

Basic reporting

The emergence of the novel coronavirus pneumonia (Covid-19) pandemic at the end of 2019 led to worldwide chaos. However, the world breathed a sigh of relief when a few countries announced the development of a vaccine and gradually began to distribute it. Nevertheless, the emergence of another wave of this pandemic returned us to the starting point. At present, early detection of infected people is the paramount concern of both specialists and health researchers.

Experimental design

This paper proposes a method to detect infected patients through chest x-ray images by using the large dataset available online for Covid-19 (COVIDx), which consists of 2128 X-ray images of Covid-19 cases, 8066 normal cases, and 5575 cases of pneumonia. A hybrid algorithm is applied to improve image quality before undertaking neural network training. This algorithm combines two different noise-reduction filters in the image, followed by a contrast enhancement algorithm. To detect Covid-19, we propose a novel convolution neural network (CNN) architecture called KL-MOB (Covid-19 detection network based on the MobileNet structure).

Validity of the findings

The performance of KL-MOB is boosted by adding the Kullback–Leibler (KL) divergence loss function when trained from scratch. The KL divergence loss function is adopted for content-based image retrieval and fine-grained classification to improve the quality of image representation. The results are impressive: the overall benchmark accuracy, sensitivity, specificity, and precision are 98.7%, 98.32%, 98.82%, and 98.37%, respectively.

Additional comments

(i) Pls give the reasons why “patients can have a good chance of survival if they are diagnosed sufficiently early.”?
(ii) Why do we need noise reduction here in “One way around these issues is to use proper image preprocessing techniques for noise reduction and contrast enhancement.”?
(iii) Some COVID-19 papers could be discussed, see “MIDCAN: A multiple input deep convolutional attention network for Covid-19 diagnosis based on chest CT and chest X-ray” and “PSSPNN: PatchShuffle stochastic pooling neural network for an explainable diagnosis of COVID-19 with multiple-way data augmentation”
(iv) Eq.2 what does Z mean?
(v) How do you design the model?
(vi) What is the effect of KL divergence?
(vii) What can you find from “In contrast, the output vector 256 in the original data achieved the best value with an accuracy of 93.24 %.”?

---

## Round 0.3 · accepted · Accept

· Academic Editor

Accept

The authors correctly addressed the points raised by the reviewer at the previous round. The article can be considered for publication.